# New Frontiers in Monoclonal Antibodies for Relapsed/Refractory Diffuse Large B-Cell Lymphoma

**DOI:** 10.3390/cancers16010187

**Published:** 2023-12-30

**Authors:** Mattia Schipani, Giulia Maria Rivolta, Gloria Margiotta-Casaluci, Abdurraouf Mokhtar Mahmoud, Wael Al Essa, Gianluca Gaidano, Riccardo Bruna

**Affiliations:** Division of Hematology, Department of Translational Medicine, University of Eastern Piedmont, Azienda Ospedaliero-Universitaria Maggiore della Carità, 28100 Novara, Italy; mattia.schipani@uniupo.it (M.S.); 20042325@studenti.uniupo.it (G.M.R.); gloria.margiotta@med.uniupo.it (G.M.-C.); abdurraouf.mahmoud@uniupo.it (A.M.M.); wael.alessa@uniupo.it (W.A.E.); riccardo.bruna@uniupo.it (R.B.)

**Keywords:** diffuse large B-cell lymphoma, monoclonal antibodies, target therapy, bispecific antibodies, antibody-dependent cellular cytotoxicity, immune checkpoint inhibitors

## Abstract

**Simple Summary:**

Forty percent of patients with diffuse large B-cell lymphoma (DLBCL) have a refractory or relapsed (R/R) disease. In this setting, prognosis is poor, particularly for patients not eligible for autologous stem-cell transplantation or CAR-T-cell therapy, thus representing an unmet need in the field of hematological malignancies. Currently, the optimal treatment approach for these patients remains controversial. Over the last few decades, monoclonal antibodies (mAbs) have dramatically changed the therapeutic landscape for cancer patients. The aim of our review is focused on novel and emerging therapeutic strategies based on different types of mAbs, including monospecific and bispecific mAbs as well as antibody–drug conjugates and immune checkpoint inhibitors, in the challenging setting of R/R DLBCL.

**Abstract:**

Diffuse large B-cell lymphoma (DLBCL) is the most common aggressive lymphoma. Approximately 60% of patients are cured with R-CHOP as a frontline treatment, while the remaining patients experience primary refractory or relapsed disease (R/R). The prognosis for R/R DLBCL patients who are neither eligible for autologous stem-cell transplantations nor CAR-T-cell treatment is poor, representing an important unmet need. Monoclonal antibodies (mAbs) have dramatically improved therapeutic options in anti-cancer strategies, offering new opportunities to overcome chemo-refractoriness in this challenging disease, even in cases of primary non-responder DLBCL. Several novel mAbs, characterized by different mechanisms of action and targets, are now available for R/R DLBCL. Unbound mAbs induce an immune response against cancer cells, triggering different mechanisms, including antibody-dependent cellular cytotoxicity (ADCC), activation of antibody-dependent cell-mediated phagocytosis (ADCP) and complement-dependent cytotoxicity (CDC). Antibody–drug conjugates (ADCs) and radioimmunotherapy (RIT), respectively, deliver a cytotoxic payload or a beta-emitter radionuclide to the targeted cells and nearby bystanders. Bispecific T-cell engagers (BiTes) and immune checkpoint inhibitors (ICIs) redirect and enhance the immune response against tumor cells. Here, we review therapeutic strategies based on monoclonal antibodies for R/R DLBCL.

## 1. Introduction

Diffuse large B-cell lymphoma (DLBCL) is an aggressive lymphoma and the most common subtype of non-Hodgkin’s lymphoma (NHL), accounting for approximately 30–40% of all diagnoses of NHL. The prevalence is higher in low/middle-income countries (~42.5%) than in high-income countries (~28.9%), with a median age at diagnosis of 53 and 65 years, respectively [1,2,3]. Most diagnoses are found in previously healthy patients, without any history of hematological malignancies, though a fraction of cases may arise from the transformation of a previous low-grade B-cell lymphoma.

The introduction of the anti-CD20 monoclonal antibody (mAb) rituximab in the CHOP regimen (R-CHOP) significantly improved the cure rate to 60% [4,5]. No other regimen, including intensified chemoimmunotherapy [6,7,8], use of second-generation anti-CD20 mAb [9], maintenance therapy [10,11,12] or targeted drugs [13,14,15,16,17], proved to be superior to R-CHOP in terms of overall survival (OS) and progression-free survival (PFS). Recently, pola-R-CHP, in which vincristine was replaced with the anti-CD79b mAb polatuzumab vedotin, has been proven to lower the risk of disease progression in previously untreated DLBCL [18].

Forty percent of patients have a refractory or relapsed disease (R/R DLBCL). Primary refractory disease (10–15% of cases) is defined as a lack of response to treatment, whereas relapsed disease (20–30% of cases), which usually occurs within the first 2 years from the end of treatment, is defined by the appearance of new lesions after achieving a complete response [19,20]. The prognosis for R/R DLBCL is very poor, particularly for primary refractory disease or early relapse (relapse within 3 to 6 months), in which the median OS is approximately 6 months [21]. The salvage regimen in transplantable-eligible patients relies on a rituximab-based chemoimmunotherapy followed by autologous stem-cell transplantation (ASCT). However, only 60% of these patients obtain a sustained remission with a 4-year progression-free survival (PFS) and OS, respectively, of 56% and 65% and an event-free survival (EFS) of 30% [22]. Most R/R DLBCL patients, however, are not ASCT eligible due to age, comorbidities or an inadequate response to salvage chemoimmunotherapy.

During the last few years, the advent of new treatments has provided alternatives to conventional therapies, in many cases obtaining sustained remission and survival improvements. The aim of this review is to provide a focus on novel mAbs in the setting of R/R DLBCL (Figure 1).

## 2. Monoclonal Antibodies

During the last few decades, numerous advances in molecular biology and in biotechnologies have significantly improved both the available number of anti-cancer drugs and therapeutic strategies, according to Paul Ehlrich’s forward-looking concept of “magic bullet” formulated more than one hundred years ago. Such a concept became available particularly thanks to the introduction of hybridoma technology [23], which provided the knowledge needed to develop mAbs capable of high selectivity towards their antigens.

Therapeutic mAbs, either chimeric or humanized, generally belong to the γ-immunoglobulin (IgG) isotype. The binding of the unbound antibody to its antigen induces an immune response against the targeted cell through antibody-dependent cellular cytotoxicity (ADCC), the activation of antibody-dependent cell-mediated phagocytosis (ADCP) and complement-dependent cytotoxicity (CDC). Another strategy relies on mAbs linked to cytotoxic drugs (antibody–drug conjugates, ADCs) or radionuclides (radioimmunotherapy, RIT) that, once bound to their target, selectively deliver their payload to cancer cells [24]. Bispecific T-cell engagers (BiTes) are mAbs capable of recognizing a tumor-associated antigen and the invariant component of the T-cell receptor (TCR) complex, CD3, redirecting a selective T-cell response against tumor cells [25]. Cancer cells can hijack immunological surveillance by overexpressing immunoinhibitory molecules (e.g., PD-L1). The blockade of these molecules or their receptors on cytotoxic lymphocytes reactivates T cells against cancer cells. This is the mechanism of action of mAbs known as immune checkpoint inhibitors [26]. Thus, both BiTes and immune checkpoint inhibitors enhance and redirect the immune response against cancer cells. Many of these mAbs have provided proof of efficacy in R/R DLBCL and have received FDA and/or EMA approval, and several others are currently under investigation or development.

## 3. Unbound Monoclonal Antibodies

Tafasitamab is an Fc-enhanced, humanized mAb that targets CD19, an antigen broadly and homogenously expressed by B-cell malignancies, including DLBCL. The Fc-enhancement consists of the introduction of two amino acid substitutions (S239D and I332E) within the Fc region that increase the affinity for Fcγ receptors on immune effector cells. The binding of tafasitamab to the CD19 antigen activates ADCC and ADCP and leads to tumor cell killing [27,28].

One of the first studies to evaluate the tolerability and efficacy of tafasitamab enrolled patients with R/R chronic lymphocytic leukemia (CLL) and demonstrated a safe profile and initial proof of clinical activity [29]. Afterwards, Jurckzac et al. evaluated tafasitamab as monotherapy in B-NHLs, highlighting the objective responses in R/R DLBCL and indolent lymphomas (26% and 29%, respectively), including rituximab-refractory diseases. Tafasitamab was well tolerated, with neutropenia as the most common adverse event (AE), especially in the heavily pretreated subgroup of DLBCL, and few grade 3 or 4 non-hematological AEs (mainly dyspnea and pneumonia) [27]. Preclinical data have shown a potential synergy of tafasitamab and lenalidomide [27], an immunomodulatory, antineoplastic and antiangiogenic agent that stimulates the proliferation and activation of NK cells, thus enhancing ADCC [30]. Lenalidomide, though, has limited clinical activity in R/R NHLs as a single agent [31,32,33,34], with a range of objective responses from 28% to 35% [28]. L-MIND, a multicenter, open-label, single-arm, phase 2 study, investigated the activity and safety of the lenalidomide and tafasitamab combination in R/R DLBCL ineligible for ASCT. The ORR was 60% (CR 43%), with a median duration of response (DOR) of 21.7 months and an 18-month PFS of 46%. The most common grade ≥ 3 AEs were thrombocytopenia, febrile neutropenia, leukopenia, anemia and pneumonia, while non-hematological AEs were of low grades (diarrhea and rash). Importantly, the incidence and severity of treatment-emergent AEs decreased after the discontinuation of lenalidomide, suggesting a main role in toxicity [28]. The long-term outcomes of the updated analysis with more than 35 months of follow-up were consistent in terms of tolerability and efficacy, with an ORR of 57.5% (CR 40%, PR 17.5%), an mDOR of 43.9 months, an mOS of 33.5 months and an mPFS of 11.6 months. The mPFS and mOS were shorter in primary refractory patients and in patients with an IPI score ≥ 3 (intermediate–high or high risk) [35]. Based on these data, tafasitamab in combination with lenalidomide was granted accelerated approval by the FDA in 2020 and conditional/accelerated approval by the EMA in 2021. Further data on patients who received treatment for ≥2 years and those in follow-up for ≥5 years confirmed that this combination provides durable responses with a decreased incidence of AEs during tafasitamab monotherapy [36]. The firmMIND trial is an ongoing, phase 3, post-authorization commitment study with the aim to confirm the efficacy, safety and overall benefit/risk of the combination in R/R DLBCL [37].

Following the approval by the FDA and EMA, many combination and comparison trials on tafasitamab in DLBCL have been performed. RE-MIND2 is a real-world study comparing the efficacy of tafasitamab plus lenalidomide (from L-MIND trial) versus other systemic anti-DLBCL regimens, including polatuzumab vedotin plus bendamustine-rituximab (pola-BR), rituximab-lenalidomide (R2) and CAR-T therapies, using a retrospective, observational, matched-cohort approach. This analysis was based on propensity score matching, which extrapolated and compared comparable populations from two different trials. Considering the limitations of this statistical approach, a significant difference in OS, favoring tafasitamab plus lenalidomide over the pola-BR and R2 cohorts, was observed, while OS appeared comparable to that of the CAR-T cohort [38]. Via matching-adjusted indirect comparisons (MAICs) of tafasitamab + lenalidomide using data from L-MIND, tafasitamab + lenalidomide was associated with a significantly improved OS, PFS, DOR and CR compared to bendamustine-rituximab (BR) [39]. B-MIND is a phase 2/3, randomized, multicenter study focused on the administration of tafasitamab with bendamustine versus BR in R/R DLBCL not eligible for ASCT (NCT02763319), whose results are ongoing. The combination of tafasitamab with different novel agents, such as plamotamab (NCT05328102) or selinexor (NCT04607772), is currently under investigation. Finally, tafasitamab plus lenalidomide is being tested as a frontline treatment in newly diagnosed DLBCL. The firstMIND and frontMIND trials explore the comparison and association of tafasitamab plus lenalidomide and R-CHOP as the first line in DLBCL, based on the treatment strategy concept that targeting both CD19 and CD20 on B tumor cells may limit target evasion in patients with low CD20 expression (Table 1) [40,41]. In conclusion, the combination of tafasitamab and lenalidomide appears to be a successful immunotherapy option in a population of heavily pretreated patients that, until recently, lacked a valid alternative to palliative treatment. A head-to-head comparison with the other novel therapies (such as CAR-T and other types of mAbs) is needed to understand the differences in terms of survival and toxicity.

## 4. Bispecific Antibodies

Bispecific antibodies are emerging as a promising off-the-shelf option for R/R DLBCL. These mAbs bind both to tumor cells and to T cells, redirecting a selective cytotoxic response against tumor cells [25].

Epcoritamab (GEN3013) is a bispecific IgG1 antibody that targets CD3 and CD20, respectively, on T and B cells, therefore activating and redirecting T cells towards B cells and causing their cytotoxicity-mediated death [42]. Epcoritamab induced effective and comparable levels of cytotoxicity in DLBCL, follicular lymphoma (FL) and mantle cell lymphoma (MCL) and it was effective in samples from newly diagnosed and R/R patients. Importantly, this evidence included patients previously exposed to other anti-CD20 mAbs, indicating that epcoritamab can be administered shortly after anti-CD20 exposure [43]. EPCORE NHL-1 (NCT03625037) was the first-in-human phase I/II trial exploring the dose escalation and dose expansion of subcutaneous epcoritamab in R/R CD20-positive NHL previously treated with an anti-CD20 mAb. In R/R DLBCL, the ORR was 88% (CR of 38%) with objective responses also achieved after CAR-T therapy [44]. No treatment-related AEs led to discontinuation or death, and all cytokine release syndrome (CRS) events were lower than grade 3 and managed with standards of care. CRSs were 49.7% (grade 1 or 2: 47.1%; grade 3: 2.5%), pyrexia 23.6%, fatigue 22.9% and ICANS 6.4% [45]. The favorable activity of a single-agent epcoritamab in such a difficult-to-treat population led to the development of further clinical trials, in monotherapy or in combinations with standard therapies, most of which are still ongoing (NCT04663347, NCT05852717, NCT04628494, NCT05578976) [46].

Mosunetuzumab is a full-length, humanized, IgG1 bispecific antibody targeting CD3 and CD20. The first-in-human dose-escalation study of a single-agent mosunetuzumab showed an ORR of 34.9% (CR 19.4%) in B-NHL. CRS was the most common toxicity in 27.4% of patients, most of which were grade 1 and 2. Other toxicities included neutropenia, hypophosphatemia, anemia and low-grade neurologic events [47]. Subsequent phase I/Ib and phase II trials documented a similar benefit–risk profile, highlighting the better rates of responses in indolent versus aggressive NHL (ORR and CR, respectively, 64.1% vs. 34.7% and 42.2% vs 18.6%), also achieving objective responses in patients previously exposed to CAR-T [48,49]. The fixed-duration regimen of mosunetuzumab could potentially reduce the cumulative safety risk and appeared to be suitable for unfit, elderly or ASCT/CAR-T-ineligible patients. Mosunetuzumab is also available in subcutaneous formulation [49,50,51], which demonstrated a clinical activity comparable to the intravenous formulation (ORR 82% and CR 64% in indolent NHL; ORR 36% and CR 20% in aggressive NHL) [51]. The combination of mosunetuzumab with polatuzumab vedotin is currently under investigation (NCT03671018) [52]. More association trials are underway, including combination with other BiTes, as glofitamab in patients who previously received CAR-T (NCT04889716); with novel ADCs, such as loncastuximab tesirine (NCT05672251); and with historical salvage platinum-based immunochemotherapy in patients who are candidates for ASCT (NCT05464329), outlining a vast scenario of possible consecutive or coexisting treatments in continuous development.

At variance from other BiTes, glofitamab is a full-length bispecific antibody with a special 2:1 configuration allowing a bivalent binding to CD20 on B cells and a monovalent binding to CD3 on T cells and potentially leading to a superior potency compared to 1:1 bispecific antibodies. Administration of the anti-CD20 obinutuzumab before glofitamab successfully reduces the risk of CRS, acting on B-cell depletion and reducing T-cell activation, with a comparable antitumor activity compared to glofitamab alone [53]. NP30179 was the first-in-human phase I trial investigating an intravenous single-agent glofitamab after single-dose obinutuzumab (7 days before) and with ongoing co-administered obinutuzumab in B-NHLs. The combination led to an ORR of 71.4% (CR 64.3%) in aggressive NHL. The most common AEs included low-grade CRS, rare self-limiting ICANS-like events and cytopenia. Importantly, the preservation of glofitamab activity observed despite the presence of a CD20 receptor competitor, such as obinutuzumab, represents a unique advantage of this antibody [54]. A phase II trial with a fixed-duration glofitamab monotherapy (12 cycles in total) showed durable CRs in R/R DLBCL, also in patients previously exposed to CAR-T cells, with 78% of complete responses at 12 months [55]. This evidence opens the question whether bispecific antibodies need to be administered until progression or for a fixed course. The numerous subsequent investigations on real-world data [56,57] and on combinations with other novel mAbs [58] will confirm the durability of the response after glofitamab and clarify its role on heavily pretreated patients, especially in patients who received prior CAR-T. Furthermore, of special interest are the ongoing trials that explore the association or the comparison of glofitamab with standards of care (NCT04408638, NCT05364424, NCT05335018) and its single-agent use in untreated DLBCL [59].

Odronextamab (REGN1979) differs from other CD3xCD20 bispecific T-cell engagers because of the minimal engineering and native antibody structure, as it is fully human, IgG4-based and hinge-stabilized [60]. The pivotal phase I first-in-human trial ELM-1 investigated the dose escalation and dose expansion of intravenous odronextamab in indolent and aggressive B-NHLs, including those with prior CAR-T therapy. Encouraging antitumor activity was observed both in FL and R/R DLBCL. The safety profile was similar to that of other BiTes, with a low rate of severe (≥grade 3) CRS and ICANS-like events. In R/R DLBCL not previously exposed to CAR-T, the ORR was 53% with all complete responses, whereas in the CAR-T-exposed population, the ORR and CR were, respectively, 33% and 27%. Interestingly, the estimated probability of maintaining a CR at 12 months was 88% in CAR-T-naïve and 100% in CAR-T-exposed patients, highlighting the potential benefit in heavily pretreated patients [61]. The subsequent phase II study ELM-2 confirmed the ELM-1 results in both CAR-T-naïve and CAR-T-exposed R/R DLBCL, with an ORR of 53% and a CR of 37%; the responses appeared to be durable with a probability of ongoing CR at 9 months of 73% [62]. Odronextamab is also under investigation in a subcutaneous formulation (NCT02290951). Furthermore, in preclinical NHL models, odronextamab has been evaluated in association with another novel antibody bispecific for CD28xCD22, REGN5837, which provides a costimulatory signal for T cells, enhancing their activation and thus promoting odronextamab activity, opening the way for a possible combination use [63].

Plamotamab is the most recent CD3xCD20 BiTe and still under investigation in a phase I dose-escalation study (NCT02924402) [64,65]. B-NHL patients, including prior CAR-T-exposed patients (50%), received intravenous single-agent plamotamab in a dose-escalation study. The preliminary results showed a manageable toxicity profile, with CRS as the most common adverse events (all of grade 1–2) and severe AEs (grade 3 or 4) consisting in anemia (19.4%), neutropenia (16.7%) and thrombocytopenia (11.1%). In DLBCL, the ORR was 47.4% (CR 26.3%) and in those prior exposed to CAR-T, the ORR was 46.2% (CR 30.8%). Based on these findings, a combination study involving tafasitamab and lenalidomide plus plamotamab versus the approved tafasitamab and lenalidomide in R/R DLBCL has just completed enrollment. The addition of plamotamab might offer a potential improved efficacy through the recruitment of distinct T-cell and Fc-mediated cytotoxic mechanisms, while also overcoming tumor resistance that might otherwise arise from the single antigenic loss of either CD19 or CD20. The results are underway (NCT05328102) (Table 2).

## 5. Antibody–Drug Conjugates

Antibody–drug conjugates (ADCs) selectively deliver cytotoxins to cancer cells, combining the specificity of mAbs with the cytotoxic effects of chemotherapy drugs. Three elements are fundamental in an ADC structure: the mAb, the covalent linker and the payload. Once attached to its cell-surface antigen, an ADC is internalized, delivering the payload to cancer cells [66]. Cytotoxins may also be released in the extracellular space either from targeted cell apoptosis or monoclonal antibody dissociation, potentially causing bystander non-targeted or antigen-negative tumor cell death, thus improving ADC efficiency [67]. The cytotoxic payloads validated in the clinical setting are categorized into two groups: DNA-interacting agents (e.g., calicheamicin, doxorubicin, pyrrolobenzodiazepine dimer) and tubulin inhibitors (e.g., maytansines and auristatins) [68].

Polatuzumab vedotin is an anti-CD79b mAb conjugated with the microtubule inhibitor monomethyl auristatin E (MMAE). CD79b is involved in B-cell receptor (BCR) signaling and widely expressed in the B-cell lineage [69]. In R/R DLBCL, polatuzumab vedotin showed modest activity as monotherapy (ORR 56%, CR 16%, mPFS 5.0 months, mDOR 5.2 months) [70] and in association with rituximab (ORR 54%, CR 21%, mPFS 5.6 months, mDOR 13.4 months) [71]. The GO29365 study evaluated pola-BR in ASCT-ineligible or ASCT-failure R/R DLBCL. Compared to BR, pola-BR achieved a higher PFS that was improved irrespective of the clinical and biological characteristics [72]. The long-term follow-up data of GO29365 confirmed the persistence of a significantly improved response and survival compared to BR at a median follow-up of approximately 50 months [73]. The most common toxicities include peripheral neuropathy, anemia, neutropenia and thrombocytopenia. Based on the GO29365 study, the FDA and EMA approved polatuzumab vedotin in combination with rituximab and bendamustine in R/R DLBCL who received at least two prior therapies.

Loncastuximab tesirine is a humanized anti-CD19 antibody conjugated with SG3199, a DNA cytotoxic agent [74]. CD19 is a pan-B marker highly expressed throughout B-cell development and maturation, until the stage of plasma cells, when CD19 expression is lost [75]. The LOTIS-2 study investigated loncastuximab tesirine as monotherapy in heavily pretreated R/R DLBCL, showing a significant antitumor activity. The objective response rate and complete response were 48.3% and 24.1%, respectively; the median DOR, PFS and OS were 10.3, 4.9 and 9.9 months [76]. Major AEs emerging during phase 1 dose escalation included neutropenia, thrombocytopenia, edema, liver test abnormalities, hypokalemia and skin/nail toxicity [77]; however, after the introduction of dexamethasone premedication, standard spironolactone diuretics and more stringent recommendations on sun exposure during a phase 2 study, pyrrolobenzodiazepine-related toxicities were generally mild-to-moderate, reversible and manageable [76]. In April 2021, the FDA and EMA approved loncastuximab tesirine for the treatment of R/R large B-cell lymphoma.

Brentuximab vedotin is an anti-CD30 antibody conjugated to monomethyl auristatin E, a microtubule-disrupting agent [78]. CD30 is expressed on a small subset of B and T cells in normal tissues, while its expression is typically high in classical Hodgkin lymphoma (cHL), anaplastic large-cell lymphoma (ALCL), cutaneous T-cell lymphoma (CTCL) and primary mediastinal B-cell lymphoma (PMBCL) [68]. Brentuximab vedotin as monotherapy in R/R DLBCL has been investigated in a phase 2 trial, showing an ORR of 44% (CR 17%, PR 27%) and a median PFS and DOR of 4 and 5.6 months, respectively. No correlation between the CD30 expression level and response was demonstrated, highlighting a plausible bystander killing effect [79]. A phase 1/dose-expansion study assessed the efficacy of brentuximab vedotin in combination with lenalidomide in R/R DLBCL. The ORR was 57% (CR 35%, PR 22%), mPFS 10.2 months and mDOR 13.1 months. Based on these results, a phase 3 study of brentuximab in combination with lenalidomide and rituximab has been initiated (ECHELON-3, NCT04404283) [80]. The most common AEs of brentuximab vedotin include fatigue, gastrointestinal disorders (e.g., diarrhea, nausea), neutropenia, pyrexia and peripheral sensory neuropathy [79].

Pinatuzumab vedotin is an anti-CD22 monoclonal antibody linked to the monomethyl auristatin E (MMAE) [81]. CD22 is involved in BCR signaling and widely expressed in more than 95% B-NHL [82], representing a potential target for the treatment of B-NHL. Pinatuzumab vedotin as monotherapy achieved an ORR of 36% (CR 16%, PR 20%) with a median PFS and DOR of 4.0 and 3.0 months, respectively. The ORR, median PFS and DOR were improved to 60% (CR 26%, PR 33%), 5.4 months and 6.2 months in combination with rituximab, as shown in the ROMULUS study. The most common adverse events included neutropenia, fatigue, peripheral sensory neuropathy, hyperglycemia and anemia [71,81].

Inotuzumab ozogamicin is a humanized anti-CD22 antibody conjugated to calicheamicin. Its antitumor activity has been studied in combination with rituximab, showing encouraging results in R/R DLBCL and an acceptable safety profile. The ORR was 74% (CR 50%), mPFS 17.1 and mDOR 17.7 months. Common toxicities included thrombocytopenia, neutropenia, hyperbilirubinemia and liver test abnormalities [83]. This combination has been investigated as an induction treatment followed by ASCT in R/R DLBCL, with unsatisfactory results [84]. In a phase 1 study, inotuzumab in combination with R-GDP achieved an ORR of 33% (CR 19%) with manageable toxicity, although gemcitabine and cisplatin doses were lower than in the standard R-GDP regimen due to hematologic toxicity [85].

Newly developed ADCs are under investigation in clinical trials of R/R B-NHLs (including DLBCL) (Table 3). Some have failed to achieve significant antitumor activity results, others have demonstrated encouraging results and several others are currently being tested [68].

## 6. Radioimmunotherapy

Radioimmunotherapy (RIT) consists of a mAb conjugated to a radionuclide that, like ADC, delivers β or γ-particles to targeted tumor cells as well as nearby tumor cells, limiting the dose of radiation delivered to normal tissues. Bystander killing allows for the eradication of tumor cells that are not necessarily targeted by the mAb [86].

Ibritumomab tiuxetan consists of ^90^yttrium, a pure beta-emitter radioisotope, conjugated to the anti-CD20 ibritumomab. The efficacy of ibritumomab tiuxetan was evaluated in R/R DLBCL in a prospective, multicenter, non-randomized phase 2 study for elderly patients. R/R DLBCL previously treated with a rituximab-based regimen exhibited lower responses and survivals (mPFS 1.6 months, mOS 4.6 months) compared to R/R DLBCL previously treated with chemotherapy only [87]. As expected, responses and survival were higher in patients without prior rituximab exposure compared to rituximab-exposed patients.

Ibritumomab tiuxetan combined with high-dose BEAM (Z-BEAM) and ASCT in R/R DLBCL was demonstrated to be efficient and safe without any additional toxicities compared to BEAM alone (2 years PFS 59% vs 30%), although the PFS difference was not statistically significant (*p* = 0.2) [88]. A recent meta-analysis showed that patients receiving Z-BEAM have a better OS (84.5%) and PFS (67.2%) at 2 years (*p* < 0.01 and *p* < 0.05) than those receiving BEAM. These data need to be further assessed with an appropriate prospective, multicenter and randomized study.

## 7. Immune Checkpoint Inhibitors

The immune system is frequently impaired in patients with hematological malignancies, both in treatment-naïve and in treated patients. Additionally, cancer cells may develop mutations capable of eluding immune checkpoints [89]. Molecular alterations involved in immunosurveillance escape are frequently observed in DLBCL and include (i) the lack of MHC class I expression (~60% of cases) due to genetic and/or epigenetic lesions in components of the HLA-I complex (e.g., β2-microglobulin, CD58 and HLA-loci); (ii) the downregulation of MHC class II (40–50% of cases), which hampers the presentation of potentially tumor-specific antigens to T cells; (iii) the overexpression of PD-L1 and PD-L2 (~20% of cases) through copy number gain, amplification and chromosomic translocation; and (iv) the increased expression of CD47 on the cancer cell surface [90,91,92]. These alterations cooperate to impair adaptive and innate immune activity against lymphoma. Immune checkpoint inhibitors (ICIs) may restore an appropriate host immune response against lymphomatous cells, potentially inducing a remission of the disease.

Normally, programmed cell death protein 1 (PD-1) and its ligands PD-L1/PD-L2 are essential for maintaining a self-tolerance environment and preventing autoimmunity [93]. However, the overexpression of PD-L1/PD-L2 by cancer cells or nonmalignant cells in the tumor microenvironment, such as tumor-associated macrophages (TAMs), hides malignant cells from the immune system. This immune escape mechanism is frequently observed in lymphomas, like classic Hodgkin lymphoma and primary mediastinal large B-cell lymphoma (PMBCL) [26,94], and to a lesser extent in DLBCL [91]. In fact, the blockade of PD-1 or PD-L1/PD-L2 enhances the immune response against lymphomatous cells. On these bases, several clinical trials have investigated the role of anti-PD-1/PD-L1 mAbs in B-cell malignancies in the relapsed/refractory setting, particularly pembrolizumab and nivolumab, which are anti-PD-1 mAbs, and atezolizumab, an anti-PD-L1 mAb. Although encouraging results have been observed in R/R cHL and R/R PMBCL, R/R DLBCL seem to be resistant to the blockade of PD-1 signaling, achieving only an ORR of 3–10% with nivolumab monotherapy [95]. These results may be partially explained by the low rate of PD-L1 overexpression in DLBCL. Nevertheless, the response to ICIs does not rely merely on the extent of PD-L1 expression, but it is a complex phenomenon involving both tumor and host characteristics. Combination regimens of anti-PD-1/PD-L1 mAbs with novel drugs (e.g., pomalidomide, mosunetuzumab) may achieve higher responses and are currently under investigation in clinical trials (Table 4).

Enhancing the adaptive immune response is not the sole mechanism to be triggered against cancer cells. Macrophages play a critical role in both innate and adaptive immunity. However, their function is frequently subverted in the tumor microenvironment, where they assume a phenotype, called M2, characterized by immunosuppressive and pro-tumorigenic properties [96]. Particularly, the overexpression of CD47 on the cancer cell surface and its interaction with signal regulatory protein α (SIRPα) on macrophages inhibit the phagocytic functions of macrophages through a so-called “do not eat me” signal [97]. Anti-CD47 mAbs block this interaction, revert macrophage phagocytic activity and induce T-cell cytotoxic response via the cross-presentations of tumor antigens on the MHC class I complex [98,99]. In a phase 1b clinical trial, magrolimab, an anti-CD47 mAb, in combination with rituximab achieved an ORR of 40% (CR 33%, PR 7%) among patients with R/R DLBCL [100]. The combination of magrolimab with rituximab and acalabrutinib, a BTK inhibitor, was investigated in a small cohort of R/R DLBCL, achieving an ORR of 28.6% (CR 28.6%) [101]. Higher responses (ORR 51.5%, CR 39.4%; mPFS 3.9 months, mDOR 18.0 months) were achieved in a phase 1b study with the combination of magrolimab and rituximab with chemotherapy [102]. Fatigue, chills, headache, anemia and infusion-related reactions were the most common treatment toxicities, and they were mostly grade 1 or 2 [100].

## 8. Conclusions

Primary refractory/relapsed DLBCL remains a challenging disease, particularly in those patients who are not eligible for ASCT/CAR-T-cell therapy, failed these treatments or experienced more than one relapse.

The landscape of anti-DLBCL treatment is evolving, and only recently, the replacement of vincristine with polatuzumab vedotin in the R-CHOP—the pola-R-CHP regimen—has demonstrated higher results as a frontline treatment [18]. In recent years, many innovative mAbs and therapeutic approaches have been developed, along with new targets, to overcome chemo-refractoriness and achieve higher response rates in the setting of R/R DLBCL. The role and survival advantages of several of these new mAbs have been well established in various lymphomas, for instance, brentuximab vedotin in cHL, systemic ALCL and in CTCL or pembrolizumab in R/R cHL. However, not all these mAbs are as effective in R/R DLBCL as they are in other lymphoma types, probably due to different biology, high disease heterogeneity, mechanisms of resistance and levels of antigen expression.

Unbound mAbs, bispecific T-cell engagers and antibody–drug conjugates provided a clinical benefit in the setting of R/R DLBCL, while the data regarding radioimmunotherapy and immune checkpoint inhibitors are less promising.

Overall, mAbs have demonstrated a tolerable safety profile. Common adverse events include hematological toxicities such as anemia, neutropenia and thrombocytopenia, as well as infusion-related reactions. Specific adverse events include peripheral neuropathy for ADCs (e.g., brentuximab vedotin, polatuzumab tesirine) and CRS and ICANS for BiTes (e.g., epcoritamab, mosunetuzumab).

Future perspectives in R/R DLBCL treatment with mAbs involve the exploration of novel targets and the development of combination regimens that can simultaneously and synergically engage more pathways, together with a personalized approach based on the molecular characteristics of the disease. Clinical trials are underway to assess the combination of mAbs with other treatments, aiming to further improve outcomes and overcome resistance. Promising results have been achieved in several clinical trials, particularly bispecific T-cell engagers are emerging as an effective and more accessible alternative to CAR-T.

Moreover, lymphomas are not the only hematological malignancies that experienced a significant shift in the treatment landscape due to the impact of mAbs. A notable example is multiple myeloma, where the anti-CD38 daratumumab has become a cornerstone in current induction regimens.

In conclusion, mAbs have represented and still represent a “game changer” of the course of several hematological malignancies, particularly of B-cell lymphomas; starting from rituximab to polatuzumab vedotin, mAbs have boldly achieved results that no other therapy has attained before in the context of lymphoma.

## Figures and Tables

**Figure 1 cancers-16-00187-f001:**
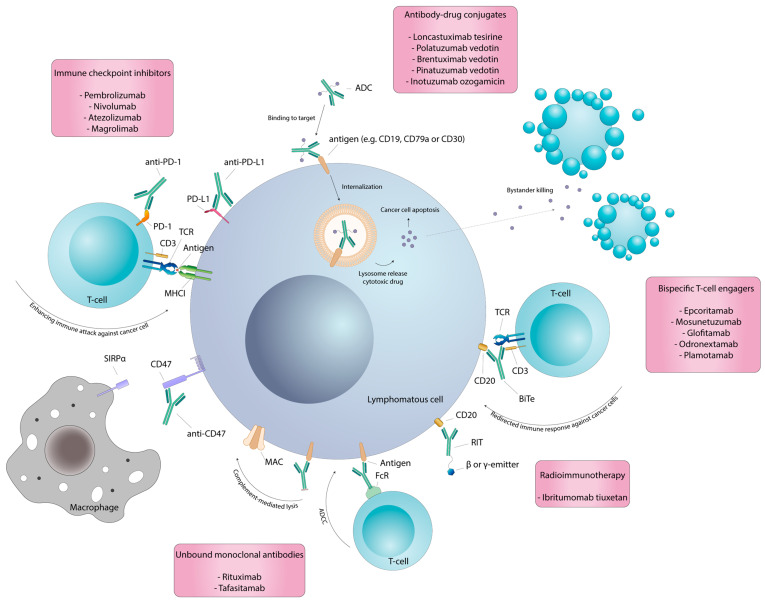
This figure illustrates the mechanisms of action of therapeutic monoclonal antibodies. The binding of an unbound monoclonal antibody to its antigen induces an immune response against targeted cancer cells through antibody-dependent cellular cytotoxicity, antibody-dependent cell-mediated phagocytosis and complement-dependent cytotoxicity. Internalization of antibody–drug conjugates into cancer cells leads to tumor cell death due to the release of cytotoxins. Following apoptosis of the targeted cancer cells and diffusion in the extracellular space, these cytotoxins can promote bystander killing. Similarly, a monoclonal antibody conjugated to a radionuclide delivers radioactive particles to targeted tumor cells as well as nearby tumor cells, resulting in their death. Blocking immune escape mechanisms, such as PD-1/PD-L1 and CD47/SIRPα signaling, restores an appropriate immune response against tumor cells. Bispecific T-cell engagers bind both tumor cells and T cells, redirecting a selective cytotoxic response against cancer cells. ADC: antibody–drug conjugates; TCR: T-cell receptor; MHCI: major histocompatibility complex class I; BiTe: bispecific T-cell engager; RIT: radioimmunotherapy; MAC: membrane attack complex; FcR: Fc receptor.

**Table 1 cancers-16-00187-t001:** Unbound monoclonal antibodies clinical trials.

NCT	Phase	Targets	Drugs	Results
NCT05626322	phase II	CD47, CD19	maplirpacept-tafasitamab-lenalidomide	NA
NCT01685008	phase IIa	CD19	tafasitamab	ORR 26%, CR 6%; mPFS 2.7 months, mDOR 20.1 months
NCT02399085	phase II	CD19	tafasitamab-lenalidomide	ORR 60%, CR 40%; mPFS 18 months in 46%, mDOR 21.7 months
NCT05429268	phase III	CD19	tafasitamab-lenalidomide	NA
NCT04697160	real world	CD19	tafasitamab-lenalidomide vs. other therapies (R2, pola-BR, CAR-T)	ORR 62.5% vs. 58.3% (pola-BR); 63.6% vs. 30.3% (R2); 59.5% vs. 75.7% (CAR-T)
NCT02763319	phase II/III	CD19	tafasitamab-bendamustine vs. rituximab-bendamustine	NA
NCT05328102	phase II	CD19, CD20/CD3	tafasitamab-lenalidomide-plamotamab vs. tafasitamab-lenalidomide	NA
NCT04607772	phase I/II	CD19, XPO1	selinexor-tafasitamab-lenalidomide vs. selinexor-other therapies	NA

R2: lenalidomide-rituximab; pola-BR: polatuzumab-bendamustine-rituximab; CAR-T: chimeric antigen receptor-T cells; ORR: overall response rate; CR: complete response; mPFS: median progression-free survival; mDOR: median duration of response; NA: not available.

**Table 2 cancers-16-00187-t002:** Bispecific antibodies clinical trials.

NCT	Phase	Target	Drug	Results
NCT03625037	phase I/II	CD20/CD3	epcoritamab	ORR 88%, CR 38% Expansion cohort: ORR 63.1%, CR 38.9%
NCT04663347	phase Ib/2	CD20/CD3	epcoritamab + other therapies	NA
NCT05852717	phase II	CD20/CD3	epcoritamab + GDP	NA
NCT04628494	phase III	CD20/CD3	epcoritamab vs. investigator choice therapy	NA
NCT02500407	phase I/Ib	CD20/CD3	mosunetuzumab	ORR 34.9%, CR 19.4%
	phase I/II	CD20/CD3	mosunetuzumab	ORR 42%, CR 23.9%; mPFS 3.2 months
	phase I/II	CD20/CD3	subcutaneous mosunetuzumab	ORR 36%, CR 20%
NCT03671018	phase Ib	CD20/CD3, CD79b	mosunetuzumab-polatuzumab	ORR: ≥65 years 72% vs. <65 years 54%; CR: ≥65 years 56% vs. <65 years 38%
NCT04889716	phase II	CD20/CD3, CD19	CAR-T followed by mosunetuzumab or glofitamab	NA
NCT05672251	phase II	CD20/CD3, CD19	mosunetuzumab-loncastuximab	NA
NCT05464329	phase Ib	CD20/CD3	DHAP or ICE + mosunetuzumab and ASCT	NA
NCT03075696	phase I	CD20/CD3, CD20	glofitamab-obinutuzumab	ORR 71.4%, CR 64.3%
	phase II	CD20/CD3, CD20	glofitamab-obinutuzumab fixed duration	ORR 95%, CR 39%; estimated 12 months OS 50%
NCT04408638	phase III	CD20/CD3, CD20	glofitamab-GemOx vs. rituximab-GemOx	NA
NCT05364424	phase Ib	CD20/CD3, CD20	glofitamab -R-ICE	NA
NCT05335018	phase II	CD20/CD3, BTK	glofitamab-poseltinib-lenalidomide	ORR 100%, CR 50%, PR 50%; mDOR 4 months
NCT02290951	phase I	CD20/CD3	odronextamab	not CAR-T exposed: ORR and CR 53%; CAR-T exposed: ORR 33%, CR 27%; mDOR NR
NCT03888105	phase II	CD20/CD3	odronextamab	ORR 53%, CR 37%; mDOR NR
NCT02924402	phase I	CD20/CD3	plamotamab	ORR 47.4%, CR 26.3%
NCT05328102	phase II	CD20/CD3, CD19	plamotamab-tafasitamab-lenalidomide	NA

GDP: gemcitabine-dexamethasone-cisplatin; CAR-T: chimeric antigen receptor-T cells; DHAP: dexamethasone-cytarabine-platinum; ICE: ifosfamide-carboplatin-etoposide; ASCT: autologous stem-cell transplant; GemOx: gemcitabine-oxaliplatin; R-ICE: rituximab-ifosfamide-carboplatin-etoposide; ORR: overall response rate; CR: complete response; mPFS: median progression-free survival; OS: overall survival; mDOR: median duration of response; NA: not available; NR: not reached.

**Table 3 cancers-16-00187-t003:** Antibody–drug conjugates clinical trials.

NCT	Phase	Targets	Drugs	Results
NCT01290549	phase I	CD79b	polatuzumab vedotin	ORR 56%, CR 16%; mPFS 5.0 months, mDOR 5.2 months
NCT01209130	phase I	CD79b	pinatuzumab vedotin	ORR 36%, CR 16%; mPFS 4.0 months, mDOR 3.0 months
NCT01691898	phase II	CD79b, CD20	rituximab-polatuzumab	ORR 54%, CR 21%; mPFS 5.6 months, mDOR 13.4 months
			rituximab-pinatuzumab	ORR 60%, CR 26%; mPFS 5.4 months, mDOR 6.2 months
NCT02257567	phase Ib/II	CD79b, CD20	polatuzumab-BR	ORR 45%, CR 40%; mPFS 9.5 months, mDOR 12.6 months
NCT04665765	phase II	CD79b, CD20	polatuzumab-R-ICE	NA
NCT02611323	phase Ib/II	CD79b, CD20	polatuzumab-obinutuzumab-venetoclax	NA
			polatuzumab-rituximab-venetoclax	NA
NCT02729896	phase Ib	CD79b, CD20, PD-L1	rituximab-atezolizumab-polatuzumab	ORR 25%, CR 13%
NCT03589469	phase II	CD19	loncastuximab tesirine	ORR 48.3%, CR 24.1%; mPFS 4.9 months, mDOR 10.3 months
NCT04384484	phase III	CD19, CD20	loncastuximab-rituximab	ORR 75%, CR 40%, PR 35%
NCT03685344	phase I	CD19, PD-L1	loncastuximab-durvalumab	NA
NCT01421667	phase II	CD30	brentuximab vedotin	ORR 44%, CR 17%; mPFS 4.0 months, mDOR 5.6 months
NCT02086604	phase I	CD30	BV-lenalidomide	ORR 57%, CR 35%; mPFS 10.2 months, mDOR 13.1 months
NCT04404283	phase III	CD30, CD20	rituximab-BV-lenalidomide	ORR 70%, CR 4 patients, PR 2 patients
NCT00299494	phase I/II	CD22, CD20	rituximab-inotuzumab	ORR 74%, CR 50%; mPFS 17.1 months, mDOR 17.7 months

BR: bendamustine-rituximab; R-ICE: rituximab-ifosfamide-carboplatin-etoposide; BV: brentuximab vedotin; ORR: overall response rate; CR: complete response; mPFS: median progression-free survival; mDOR: median duration of response; NA: not available.

**Table 4 cancers-16-00187-t004:** Immune checkpoint inhibitors clinical trials.

NCT	Phase	Targets	Drugs	Results
NCT02038933	phase II	PD-1	nivolumab	ORR 3–10%, CR 0–3%; mDOR 8–11.4 months
NCT02953509	phase Ib	CD47, CD20	magrolimab-rituximab	ORR 40%, CR 33%
NCT03527147	phase I	CD47, CD20, BTK	magrolimab-rituximab-acalabrutinib	ORR 28.6%, CR 28.6%
NCT02953509	phase Ib	CD47, CD20	magrolimab-rituximab-GemOx	ORR 51.5%, CR 39.4%; mPFS 3.9 months, mDOR 18.0 months

GemOx: gemcitabine-oxaliplatin; ORR: overall response rate; CR: complete response; mPFS: median progression-free survival; mDOR: median duration of response; NA: not available.

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
