# Peer review of "New Frontiers in Monoclonal Antibodies for Relapsed/Refractory Diffuse Large B-Cell Lymphoma"

_cancers, 2023, doi:10.3390/cancers16010187_

Round 1

Reviewer 1 Report

Comments and Suggestions for Authors

Comments for Schipani et al.

The review article by Schipani et al titled “New Frontiers in Monoclonal Antibodies for Relapsed/Refrac-tory Diffuse Large B Cell Lymphoma” focused on novel and emerging therapeutic strategies based on different types of antibody-based therapies for relapsed/refractory DLBCL.

Instead of placing “NA”, I would advise updating the “Results” section in tables 2 and 3 based on the following new information released.

Table2

Trial: NCT04663347

Results: The ongoing phase 1b/2 trial evaluating epcoritamab and R2 among 28 evaluable patients with R/R FL (EPCORE NHL-2 arm 2; NCT04663347), ORR was 100%, with a complete metabolic response observed in 96%. Most cytokine release syndrome (CRS) events were of low grade (grade 1, 30%; grade 2, 13%; grade 3, 7%) and occurred in cycle 1; all CRS events resolved with standard management.

https://ashpublications.org/blood/article/140/Supplement%201/9338/490737/Phase-3-Trial-of-Subcutaneous-Epcoritamab-in

Trial: NCT05335018

Results: There was 1 event of cytokine release syndrome grade 1, which resolved without sequalae. The most common adverse event was skin rash (2 grade 2, 1 grade 1) which did not require study drug modifications. Neutropenia occurred in 2 patients, but no dose adjustment was necessary. One patient with a previous history of prior COVID19 infection after TNB-486 (CD19/CD3 bispecific) clinical trial, despite receiving vaccination prior to this study enrollment, experienced COVID19 reactivation after cycle 1. Glofitamab D8 and D15 doses were skipped, and both lenalidomide and posetinib were stopped for 14 days. The patient recovered and underwent full doses of subsequent therapies. After DSMB clearance, we are now enrolling patients in the expansion phase.

For tumor response, all 6 patients achieved objective response per Lugano criteria at the end of cycle 2 (3 CR, 3 PR). Based on the best overall response, there were 4 confirmed CR and the median duration of response 4 months

https://www.ncbi.nlm.nih.gov/pmc/articles/PMC10431110/

Trial: NCT05328102

Terminated

https://clinicaltrials.gov/study/NCT05328102

Table3

Trial: NCT04384484

Results: The median age of patients treated on the trial was 74.5 years of age (range, 35-93) and the median number prior systemic therapies was 1 (range, 1-6). Most patients were female (55%; n = 11); 45% were male (n = 9). Enrolled patients had stage II (30%; n = 6), stage III (30%; n = 6), or stage IV (40%; n = 8) disease. A majority (85%; n = 17) of patients presented with DLBCL not otherwise specified, and 15% (n = 3) of patients presented with high-grade B-cell lymphoma with MYC and BCL2 and/or BCL6 rearrangements.

All patients evaluated did not undergo previous stem cell transplant and most patients had relapsed (90%; n = 18) on their first-line therapy.

The median number of doses administered was 5 (range, 1-8).

In terms of safety, 95% (n = 19) of patients had at least 1 treatment-emergent adverse effect (TEAE), and 50% of patients (n = 10) had a grade 3 or higher TEAE.

The most common TEAEs recorded were rash (25%; n = 5), fatigue (20%; n = 4), and increased gamma-glutamyltransferase (20%; n = 4). Grade 3 or greater TEAEs included increased gamma-glutamyltransferase (15%; n = 3), increased alanine aminotransferase (10%; n = 2), and neutropenia (10%; n = 2).

https://www.onclive.com/view/loncastuximab-tesirine-rituximab-elicits-promising-anti-tumor-activity-in-relapsed-refractory-dlbcl

Trial: NCT04404283

Results: 10 pts with R/R DLBCL were enrolled. Median age was 70.5 years, 7 pts were male, and all had an ECOG ≤1. Median prior lines of therapy was 3 (range, 2 to 6); no pts received prior HSCT and 6 pts received prior CAR-T. At a median follow-up of 6.9 months (range, 2.3 to 14.1), the most common treatment emergent adverse events (TEAE) were fatigue (n = 5), anemia (n = 4) and constipation (n = 4). Grade ≥3 events were experienced by 8 pts, most commonly anemia and pneumonia (n = 3 each), and neutropenia and thrombocytopenia (n = 2 each). Serious adverse events were observed in 7 pts. Seven pts each had BV and rituximab dose modifications, and 6 pts had a len dose modification. The most common reasons for any dose modification were anemia (n = 3), neutropenia, peripheral neuropathy, or pneumonia (n = 2 each). 4 pts discontinued treatment due to an AE, 2 of which were treatment related (n = 1 each of Grade 2 fatigue; Grade 3 anemia). There was 1 death due to a TEAE that was not treatment related. The ORR (best response) was 70%, including 4 pts with a complete metabolic response, 2 pts with a partial metabolic response, and 3 pts with progressive disease. Responses were seen in both CD30 (+) and (-) pts, as well as in 4 pts who received prior CAR-T.

https://ascopubs.org/doi/abs/10.1200/JCO.2022.40.16_suppl.7559

Author Response

Thank you for your kind review. We have updated table 2 and 3 according to your suggestions.

However, the data you suggest for the clinical trial ‘NCT04663347’ refers to R/R FL and not DLBCL; therefore, according to the focus of our manuscript (R/R DLBCL), we prefer to keep ‘NA’.

Regarding the clinical trial ‘NCT05328102’, as you suggested, the study is terminated. Unfortunately, data has not yet been published (neither on https://clinicaltrials.gov/study/NCT05328102?tab=results, nor on https://www.clinicaltrialsregister.eu/ctr-search/trial/2021-003658-22/results, nor in abstracts or PubMed).

Reviewer 2 Report

Comments and Suggestions for Authors

This review deals with the current therapeutic approach of Lymphoma by the relevant use of monoclonal antibodies. I think that the use of some terms is either unnecessary (e.g. “worldwide” in the abstract) or confusing (e.g. “naked”; unbound?, raw), bare?); e.g. “activated (P5,p2,L2)”; conducted?, performed? and real world study” (P5,p2,L2); study?, trial?). 

Comments on the Quality of English Language

No comments.

Author Response

Thank you for your kind review. We have removed “worldwide” from the abstract and replaced “activated” (P5, p2, L2) with “performed” throughout the text, as you suggested.

Concerning the term ‘real-world study’ (P5, p2, L2), it specifically refers to the unique approach employed in the 'RE-MIND2' study. The control arm was derived from retrospectively collected data in a 'real-world' setting, aligning with the eligibility criteria of the study. In fact, as stated in ClinicalTrial.gov, ‘This retrospective observational cohort study aims to generate a historical control consisting of R/R DLBCL patients who received currently guideline recommended therapies.’

https://clinicaltrials.gov/study/NCT04697160?term=NCT04697160&rank=1

In addition, as per your suggestion the term “naked” has been replaced with “unbound”.

Reviewer 3 Report

Comments and Suggestions for Authors

The manuscript "New Frontiers in Monoclonal Antibodies for Relapsed/Refractory Diffuse Large B Cell Lymphoma" is a detailed review focused on emerging monoclonal antibody (mAb) therapies in the treatment of relapsed/refractory diffuse large B-cell lymphoma (R/R DLBCL). It discusses various mAb therapies, including monospecific and bispecific mAbs, antibody-drug conjugates, and immune checkpoint inhibitors, addressing their mechanisms of action and clinical applications.The authors cover a wide range of monoclonal antibodies, offering a holistic view of current and emerging therapies. They also provide an in-depth look at the mechanisms and efficacy of different mAbs, backed by clinical trial data. Overall, the manuscript is well-writen and provides a valuable resource for understanding the evolving landscape of mAb therapies in R/R DLBCL.

Nevertheless, the study has a few weaknesses:

1) The focus is on R/R DLBCL, which may limit the application of findings to other types of lymphomas or cancers. I would suggest the authors include a few comparisons or implications of these therapies in the broader context of lymphoma treatment.

2) The authors could also put more emphasis on the future trajectory of mAb research and potential areas of innovation. Please mention more information on ongoing trials and emerging research to provide a forward-looking perspective.

3) Please expand on the limitations and potential side effects of these therapies to provide a balanced view.

4) Just a few grammatical points for the authors to pay attention to (e.g., "immune cherckpoint inhibitors" in the Abstract - please correct to "immune checkpoint inhibitors")

Author Response

Thank you for your kind review. Below, you will find our point-by-point responses.

The manuscript "New Frontiers in Monoclonal Antibodies for Relapsed/Refractory Diffuse Large B Cell Lymphoma" is a detailed review focused on emerging monoclonal antibody (mAb) therapies in the treatment of relapsed/refractory diffuse large B-cell lymphoma (R/R DLBCL). It discusses various mAb therapies, including monospecific and bispecific mAbs, antibody-drug conjugates, and immune checkpoint inhibitors, addressing their mechanisms of action and clinical applications.The authors cover a wide range of monoclonal antibodies, offering a holistic view of current and emerging therapies. They also provide an in-depth look at the mechanisms and efficacy of different mAbs, backed by clinical trial data. Overall, the manuscript is well-writen and provides a valuable resource for understanding the evolving landscape of mAb therapies in R/R DLBCL.

Nevertheless, the study has a few weaknesses:

  • The focus is on R/R DLBCL, which may limit the application of findings to other types of lymphomas or cancers. I would suggest the authors include a few comparisons or implications of these therapies in the broader context of lymphoma treatment.
    We thank the Reviewer for her/his comment. Because the declared focus of the manuscript is R/R DLBCL, which is already a very vast topic, we prefer restricting our analysis to this lymphoma type, as also stated in the original manuscript title. However, a few comparisons have been included, as per your suggestion (p. 12, l. 9-11; p. 12, l. 31-34).
  • The authors could also put more emphasis on the future trajectory of mAb research and potential areas of innovation. Please mention more information on ongoing trials and emerging research to provide a forward-looking perspective.
    According to your suggestions, we have expanded our conclusions including some comparisons, implications and future perspectives of anti-lymphoma treatment (p. 12, l. 23-30).
  • Please expand on the limitations and potential side effects of these therapies to provide a balanced view.
    As suggested, we have expanded on the limitations and potential side effects of these therapies (p. 12, l. 11-22).
  • Just a few grammatical points for the authors to pay attention to (e.g., "immune cherckpoint inhibitors" in the Abstract - please correct to "immune checkpoint inhibitors").
    The manuscript was revised for grammatical and typing errors.